# Prophylactic Feeding of *Clostridium butyricum* and *Saccharomyces cerevisiae* Were Advantageous in Resisting the Adverse Effects of Heat Stress on Rumen Fermentation and Growth Performance in Goats

**DOI:** 10.3390/ani12182455

**Published:** 2022-09-16

**Authors:** Ligang Xue, Dan Wang, Fangyu Zhang, Liyuan Cai

**Affiliations:** 1College of Animal Science and Technology, Jilin Agricultural Science and Technology University, Changchun 130118, China; 2College of Veterinary Medicine, Jilin Agricultural University, Changchun 130118, China; 3Institute of Animal Husbandry and Veterinary Medicine, Jilin Academy of Agricultural Sciences, Changchun 130033, China; 4Department of Animal Nutrition and Feed Science, College of Animal Science and Technology, Huazhong Agricultural University, Wuhan 430070, China

**Keywords:** goats, prophylactic feeding, *Clostridium butyricum*, *Saccharomyces cerevisiae*, heat stress, rumen fermentation, growth performance

## Abstract

**Simple Summary:**

Heat stress occurs when goats are exposed to high environmental temperatures and humidity for a long period. Heat stress could adversely affect the rumen fermentation and growth performance of goats. Dietary supplementation with *Clostridium butyricum*, *Saccharomyces cerevisiae*, and their mixture were effective ways to alleviate the effects on the rumen fermentation and growth performance of heat-stressed goats. In this study, these two probiotics and their mixture were supplemented in goats for a period before heat stress. The results showed that these probiotics effectively alleviate the adverse effects of heat stress by promoting rumen fermentation and growth performance. Therefore, this study provides a reference for applying these two probiotics at the optimum timing to alleviate the adverse effects of heat stress on goats.

**Abstract:**

This study aimed to investigate the effect of the prophylactic feeding of *Clostridium butyricum (CB), Saccharomyces cerevisiae* (SC), and their mixture before the onset of heat stress on the rumen fermentation and growth performance of goats, and subsequently, on heat stress status. Forty-eight male Macheng Black × Boer crossed goats (22.25 ± 4.26 kg) were divided into four groups—the control group (fed the basal diet), and the CB (0.05% CB added to the basal diet), SC (0.60% SC added to the basal diet), and Mix (0.05% CB and 0.60% SC added to the basal diet) groups—and fed for fourteen days. Then, these goats were kept in a heat stress environment (with a temperature–humidity index of 87.04) for fourteen days. Then, the parameters of rumen fermentation and growth performance were measured. The results showed that the pH values, the activities of cellulolytic enzymes (avicelase, CMCaes, cellobiase, and xylanase), and the concentrations of ammonia-N, total volatile fatty acid, acetic acid, propionic acid, and butyric acid were significantly increased (*p* < 0.05) in the rumens of the CB, SC, and Mix groups compared to those of the control group. Moreover, the average daily gain and the digestibility of dry matter, neutral detergent fiber, and acid detergent fiber were significantly increased (*p* < 0.05) in the CB, SC, and Mix groups compared to those of the control group. These results suggest that these two probiotics and their mixture effectively alleviate the adverse effects of heat stress on rumen fermentation and growth performance via prophylactic feeding.

## 1. Introduction

With the prohibition of antibiotics in livestock production, probiotics are being used more widely and frequently in ruminant production to improve rumen fermentation, post-ruminal nutrient uptake, intestinal health, and performance [1,2,3]. There are three generally recognized action mechanisms for probiotics in ruminants: (1) When stress or disease cause a change in microecological balance in the gastrointestinal tract, increasing the abundance of harmful bacteria and decreasing the abundance of beneficial bacteria, thus affecting the growth performance of ruminants. When probiotics enter the gastrointestinal tract, they will compete with harmful bacteria in the gastrointestinal tract, inhibit the growth of pathogenic bacteria, and finally, form a stable and beneficial microecological environment [4]. (2) When probiotics enter the digestive system, a layer of the probiotic membrane will be formed on the gastrointestinal mucosa. This membrane can prevent many pathogenic bacteria from attaching to the gastrointestinal wall, prevent the absorption of toxic components, control harm caused by pathogens in the body, and maintain an excellent microecological balance [2]. (3) Most probiotics are anaerobic microbiota, and thus, they form an anaerobic environment. Thus, the number of pathogenic aerobic and facultative aerobic bacteria will decrease, making the beneficial flora become the dominant flora, thus promoting the growth of animals [5]. *Clostridium butyricum* is a strictly anaerobic endospore-forming Gram-positive butyric acid-producing bacterium and is a promising probiotic candidate [6,7]. *Saccharomyces cerevisiae* is a facultative anaerobe commonly used in ruminants [8]. These two probiotics all play a significant role in promoting rumen fermentation and improving growth performance [7,9,10]

Ruminants are the most heat-intolerant animals since their rumen fermentation produces a large amount of heat [11]. Heat stress has been shown to cause a decrease in the abundance of cellulolytic bacteria and increase the abundance of starch-decomposing bacteria [12]. Therefore, the rumen environment is disturbed, and the rumen fermentation capacity declines significantly; with a decrease in pH values, the concentration of ammonia-N, and volatile fatty acids (VFAs) increases in the rumen. Moreover, the digestibility of dry matter (DM), neutral detergent fiber (NDF), and acid detergent fiber (ADF) has been shown to decrease, which eventually leads to a decline in the growth performance of ruminants [13,14,15,16].

Our previous study found that a diet with *Clostridium butyricum* and *Saccharomyces cerevisiae* during the heat stress period was beneficial for the rumen fermentation and growth performance of goats [7,10,17]. However, when a hot summer is coming, goat breeders cannot accurately judge the exact time of the occurrence of heat stress. Thus, probiotics cannot be used effectively to improve rumen fermentation and promote growth. Therefore, this study aimed to investigate the effect of the prophylactic feeding of *Clostridium butyricum, Saccharomyces cerevisiae*, and their mixture before the onset of heat stress for a period of time on the rumen fermentation and growth performance of these goats, and subsequently, on heat stress status.

## 2. Materials and Methods

### 2.1. Goats and Probiotic Feeding Experiment

This study was approved by the Animal Care and Use Committee of the Jilin Agricultural Science and Technology University (Approval code: 20221011) and carried out from June to August. Throughout this study, the goats were fed twice daily at 7:00 and 17:00 with a 1.30 kg basal diet and free access to water. The composition and nutritional levels of the diet were set according to Mo [18]. The forty-eight male Macheng Black × Boer crossed goats (22.25 ± 4.26 kg) aged 5.0 ± 1.0 months were divided into four groups: the control group (fed the basal diet), and the CB (0.05% *Clostridium butyricum* of DM concentration added to the basal diet), SC (0.6% *Saccharomyces cerevisiae* of DM concentration added to the basal diet), and Mix (0.05% *Clostridium butyricum and* 0.6% *Saccharomyces cerevisiae* of DM concentration added to the basal diet) groups. The commercial *Clostridium butyricum* product was provided by Huijia Biotechnology Co., Ltd. (Huzhou, China) with a 2.0 × 10^10^ CFU/g live cell concentration. Angel Yeast Co., Ltd. (Yichang, China) provided the commercial *Saccharomyces cerevisiae* product with a 1.0 × 10^8^ CFU/g live cell concentration. During the feeding experiment, these goats were kept in a temperature- and humidity-controlled room where the temperature, relative humidity, and temperature humidity index (THI) were 25.0 ± 1.0 °C, 62.0 ± 1.5%, and 73.01 ± 2.11 for fourteen days (control period). Then, the temperature, relative humidity, and temperature humidity index (THI) of this room were altered to 33.2 ± 1.20 °C, 73.3 ± 2.3%, and 86.8 ± 0.8, respectively, to ensure that the goats were in a heat stress thermal environment for fourteen days (heat stress period). All the goat groups were fed a basal diet during the whole heat stress period. The composition and nutritional levels of the diet are given in Table 1. Five grams of Cr_2_O_3_ were added to the diet from days 11 to 13. The procedure of the experiment is given in Figure 1.

### 2.2. Sampling and Measurements

Fresh fecal samples were collected from the rectum of all goats before feeding on days 12 to 14 during the heat stress period, and these samples were stored at −20℃ for further analysis. Blood samples were collected from the jugular vein of all the goats in the morning (after 24 h of fasting) on day 14 of the heat stress period. The peripheral blood lymphocytes were isolated from whole blood using a specific kit from Solarbio Science & Technology (Beijing, China) to determine the expression levels of heat shock protein 70 (HSP 70) genes. The primer sequences, as described by Cai et al. [10] and Chaidanya et al. [19], were synthesized by Sangon Biotech Co., Ltd. (Shanghai, China). The details of these primers are given in Table 2.

TRIzol^®^ Reagent (Life Technologies, CA, USA) was used to extract total RNA from the peripheral blood lymphocytes following the manufacturer’s instructions. Reverse transcription was performed using the Revert Aid First Strand cDNA Synthesis kit (Thermo Fisher Scientific, Waltham, MA, USA) following the manufacturer’s instructions. An SYBR RT-PCR Kit (Bio-Rad, Hercules, CA, USA) in conjunction with a real-time fluorescent quantitative PCR system (Life Technologies, Carlsbad, CA, USA) was used for the RT-PCR. The conditions of PCR reaction were 94 °C for 3 min, 30 cycles of 94 °C for 30 s, 50 °C for 45 s, and 72 °C for 45 s, and a final extension at 72 °C for 10 min. Each sample was analyzed in triplicate, and the levels of relative expression were quantified using the 2^−AACt^ method [20]. For measurement of the concentration of cortisol, blood samples were centrifuged at 3000 rpm for 10 min to obtain serum. A cortisol assay kit provided by the Nanjing Jiancheng Bioengineering Institute (Nanjing, China) was used to measure the concentration of serum cortisol following the manufacturer’s instructions.

Rumen fluids were collected from all goats using a soft plastic stomach tube with a GM-0.33A vacuum pump (Jinteng, Tianjin, China) after feeding for 4 h in the morning on day 14 of the heat stress period. The values of pH and oxidation-reduction potential (ORP) were determined immediately using a pH meter and an ORP meter (Orion Technology Co. Ltd., Massachusetts, USA), respectively. Then, the rumen fluids and supernatants were collected. As described by Maitisaiyidi et al. [21], ammonia nitrogen (NH_3_-N) was measured using an ultraviolet–visible spectrophotometer (Thermo Fisher Scientific, Waltham, MA, USA). Yang et al. [22] described that volatile fatty acids (VFAs) were measured using gas chromatography. In brief, 1.0 mL of 25% (*w/v*) metaphosphoric acid was added to 0.20 mL of ruminal supernatant and centrifuged at 10,000 r/min for 10 min. Then, the liquid supernatant was injected into a 30 m × 0.53 mm × 1.00 um Chrompack CP-Wax 52 fused silica column in a gas chromatograph equipped with a Model 2010 flame ionization detector (Shimazu, Kyoto, Japan).

During the heat stress period, the amounts of feed given and left were recorded for each goat to calculate the daily matter intake (DMI), and the body weights were recorded at the start and end of this period to calculate the average daily gain (ADG). The dry matter (DM) was measured in the feed and feces according to method #930.15 in the AOAC [23], and the digestibility (%) of DM was calculated as: DM content in feedstuff—DM content in feces)/ DM content in feedstuff ×100; this was as described by Goering and Van Soest [24]. The neutral detergent fiber (NDF) and acid detergent fiber (ADF) were measured in the feed and feces. The digestibility (%) of these two parameters was calculated as: NDF or ADF content in feedstuff−NDF or ADF content in feces)/NDF or ADF content in feedstuff ×100.

### 2.3. Statistical Analysis

The data were analyzed using the “stats” package in R studio (v.4.0.2) (Allaire, Wall Township, NJ, USA). To reveal significantly different physiological parameters, gene expression levels, and cortisol concentrations between the control and heat stress periods, a two-tailed Student’s *t*-test analysis was performed. To reveal the significantly different parameters among the probiotic-supplemented groups, two-way analysis of variance (ANOVA) tests followed by a post hoc Dunn test for multiple pairwise comparisons were performed. *p* values of less than 0.05 were considered statistically significant.

## 3. Results

### 3.1. Parameters for Evaluation of the Occurrence of Heat Stress in Goats

In the heat stress period, the goats’ skin temperature, respiratory rate, and pulse were significantly increased (*p* < 0.05); however, there were no significant differences (*p* > 0.05) in the rectal temperature of goats compared to the control period. The skin temperature, rectal temperature, respiratory rate, and pulse of the goats are given in Table 3.

To further confirm that the goats were experiencing heat stress, the HSP 70 genes in the rumen fluid and blood were determined. The expression of HSP 70 and HSPA 1 was significantly increased (*p* < 0.05; Figure 2A) in rumen fluid and blood, respectively; but there were no differences in the expression of HSPA 6 and HSPA 8 in the blood of goats between the control period and the heat stress period (*p* > 0.05; Figure 2A). Additionally, the cortisol concentrations were significantly increased in the serum of goats under heat stress compared to those in the control period (*p* < 0.05; Figure 2B).

### 3.2. Prophylactic Feeding of Probiotics Improved Rumen Environment and Enhanced Rumen Fermentation of Heat-Stressed Goats

The pH values were significantly increased in the rumen of the CB, SC, and Mix groups compared with that of the control group (*p* < 0.05). Meanwhile, the ORP values were significantly increased in the rumen of the CB, SC, and Mix groups compared with that of the control group (*p* < 0.05). The concentration of NH_3_-N was significantly increased (*p* < 0.05) in the CB, SC, and Mix groups compared with that of control group. Additionally, the concentrations of TVFA, acetic acid, propionic acid, and A/P ratio were significantly increased (*p* < 0.05) in the CB, SC, and Mix groups compared with that of control group. The activities of rumen avicelase, CMCase, cellobiase, and xylanase in the CB, SC, and Mix groups were significantly increased (*p* < 0.05) compared with those in the rumen of control group. Additionally, the activities of these four rumen enzymes in the Mix group were significantly higher than those in the CB and SC groups (*p* < 0.05). The parameters of rumen fermentation and the activities of the ruminal cellulolytic enzymes of heat-stressed goats with probiotic supplements are shown in Table 4.

### 3.3. Prophylactic Feeding of Probiotics Improved Growth Performance of Heat-Stressed Goats

The DMI, ADG, and digestibility of DM, NDF, and ADF were significantly increased (*p* < 0.05) in the CB, SC, and Mix groups compared with those of the control group. Additionally, the Mix group exhibited a more significant effect in enhancing these growth performance parameters. The parameters of the growth performance of heat-stressed goats with probiotic supplements are shown in Table 5.

## 4. Discussion

In the hot summer, heat stress is an inevitable issue of intensive goat production in the Jianghuai region of China [12,16]. Heat stress has adverse effects on rumen fermentation and the growth performance of ruminants [11,16,25]. Our previous studies showed that the parameters—including the pH value and the concentrations of NH_3_-N, TVFA, acetic acid, propionic acid, butyric acid, and cellulolytic enzyme activity—was significantly decreased in heat-stressed goats. Moreover, the parameters of growth performance, including DMI, ADG, and the digestibility of DM, NDF, and ADF, were significantly decreased in heat-stressed goats [16]. Dietary supplementation with probiotics was a good way to alleviate the adverse effects of heat stress on livestock [26,27].

A prophylactic feeding strategy was applied in this study. Prophylactic feeding means supplementation with probiotics for a period of time before heat stress occurrence to investigate the effects of these probiotics on rumen fermentation and growth performance. This is closer to the practice of goat production. In this study, the pH values in the rumen were significantly increased with dietary supplementation with *Clostridium butyricum, Saccharomyces cerevisiae,* and their mixture. This result is consistent with previous studies that supplement cows with live yeast in the summer [7,10,28,29,30]. Few studies have investigated the effects of *Clostridium butyricum* on ruminal pH values. A previous study reported that supplemented calves, which were fed a 50% high-concentrate diet for one week with *Clostridium butyricum* at 1.5 or 3.0 g/100 kg body for five days, the 24 h mean ruminal pH value was increased [31]. This result suggests that these two probiotics could effectively resist the pH decrease caused by heat stress [16]. This function could be explained by the fact that that *Clostridium butyricum* and *Saccharomyces cerevisiae* could produce organic acids, vitamins, and other nutrient factors in the rumen, which could promote the activity of lactic acid-utilizing bacteria [32]. Additionally, yeast could promote the utilization of soluble sugars by gut microbiota, and then, reduce lactic acid production [30]. Therefore, these probiotics have an effect on rumen pH stabilizing. Moreover, the abundance of rumen protozoa may be enhanced by these two probiotics, which could also lower the ruminal lactic acid concentration [33]. However, previous studies also reported that supplementation with *Saccharomyces cerevisiae* has no effect on pH [34,35,36,37] or could decrease ruminal pH [36]. These differences could be due to the sources, strains, or doses, the timing of feeding of probiotics, the animal species, or their health status [12]. In this study, supplementation with *Clostridium butyricum*, *Saccharomyces cerevisiae*, and their mixture could enhance the production of VFAs, including TVFA, acetic acid, propionic acid, and butyric acid. Previous studies have shown inconsistent results regarding the effects of *Saccharomyces cerevisiae* on VFAs. Previous studies reported that supplementation with *Saccharomyces cerevisiae* or active dry yeast could significantly increase the concentration of TVFA, acetic acid, propionic acid, butyric acid, and the A/P ratio in the rumen [7,38,39,40,41,42,43]. However, a previous study has shown that supplementation with *Saccharomyces cerevisiae* has no effect on VFA concentration in the rumen [44]. Until now, few studies have reported the effects of *Clostridium butyricum* on ruminal VFA production. It was reported that *Clostridium butyricum* improves the ruminal VFA production of heat-stressed goats in vitro and in vivo [7,10]. However, in calves, supplementation with *Clostridium butyricum* at 1.5 or 3.0 g/100 kg (body weight) did not affect ruminal VFA production [36]. In this study, supplementation with *Clostridium butyricum* and its mixture with *Saccharomyces cerevisiae* could enhance the concentration of TVFA, acetic acid, propionic acid, butyric acid, and the A/P ratio. The increased trends of VFA concentration due to the effects of these two probiotics promote the activities of ruminal fibrolytic bacteria [12,42,45]. In this study, supplementation of the A/P ratio was increased compared to that of the control, suggesting that probiotics may change in fermentation mode in the rumen [16]. The increase in the A/P ratio in this study is attributed more to the increased extent of acetic acid than that of propionic acid. One opinion has been held which states that the change in VFA concentration through supplementation with probiotics is temporary; once probiotic supplementation is terminated, this rising effect disappears [7]. This problem should be taken into consideration in nutrition studies and the production practices of ruminants.

In this study, the DMI and ADG were significantly increased with dietary supplementation with *Clostridium butyricum, Saccharomyces cerevisiae,* and their mixture. This result is consistent with previous studies that state that supplementation with yeast at 0.2 g/day increased feed intake in the early lactation stage of dairy goats [46]. However, another previous study reported that there was no effect on DMI on cows in summer with *Saccharomyces cerevisiae* supplementation [28]. There are few studies that investigate the effect of *Clostridium butyricum* on DMI, especially in ruminants under heat stress. Moreover, our study shows that supplementation with *Saccharomyces cerevisiae* could increase ADG in heat-stressed goats. This result is consistent with previous studies that found that with *Saccharomyces cerevisiae* supplementation, the ADG increased both in cows and goats [7,47,48,49]. Similarly, *Clostridium butyricum* was found to have positive effects on the ADG of farm animals [7,10,50]. In this study, *Clostridium butyricum, Saccharomyces cerevisiae,* and their mixture have positive effects on the digestibility of DM, NDF, and ADF. These results were similar to previous studies that found that the digestibility of DM, NDF, and ADF in sheep and goats was improved through supplementation with *Saccharomyces cerevisiae* [7,36,44]. Previous studies scarcely investigated the effects of *Clostridium butyricum* on these parameters of digestibility. As a promising candidate for microbial additives [50], it is necessary to evaluate the effects of *Clostridium butyricum* on the digestibility of DM, NDF, and ADF. In this study, supplementation with *Clostridium butyricum* could promote the digestibility of DM, NDF, and ADF in heat-stressed goats.

These two probiotics can improve digestibility because they are good sources of vitamins, organic acids, and minerals, which promote the abundance of cellulolytic bacteria and fungi in the rumen and may improve fiber digestion [15,51].

Despite the fact that *Saccharomyces cerevisiae* and *Clostridium butyricum* can have positive effects on the growth performance of ruminants, their effects may vary in various studies. This discrepancy is due to factors such as differences in the composition of diets, the number of live cells of probiotics, supplement levels, and management strategies [51]. In this study, the activities of cellulolytic enzymes were significantly increased in probiotic-supplemented groups, which could be a reason for the improvements in the digestibility of DM, NDF, and ADF.

## 5. Conclusions

Prophylactic feeding with *Saccharomyces cerevisiae*, *Clostridium butyricum*, and their mixture could improve the rumen environment and fermentation by increasing pH values and VFA concentrations in the rumen. Moreover, the digestibility of DM, DNF, and ADF was increased by these probiotic supplements. Thereafter the growth performances were enhanced by increasing the DMI and ADG of heat-stressed goats. Therefore, in practical goat production, prophylactic supplementation with *Saccharomyces cerevisiae* and *Clostridium butyricum* can be an effective way to alleviate the adverse effects on rumen fermentation and growth performance in heat-stressed goats.

## Figures and Tables

**Figure 1 animals-12-02455-f001:**
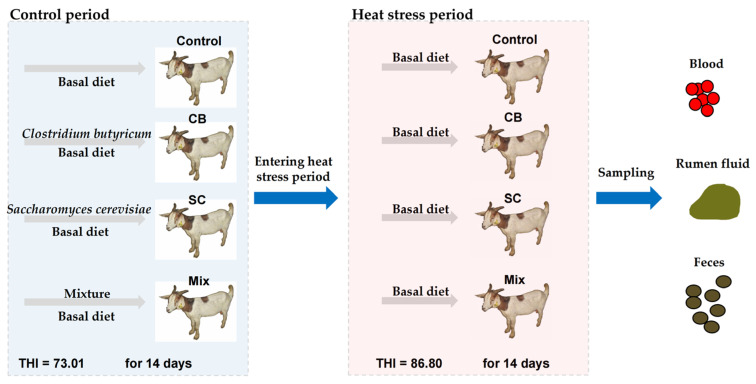
Heat stress modeling and sampling process of goats. Goats in the blue box are in the control period, where the environmental temperature humidity index (THI) was 73.01, and goats were in this environment for 14 days. Goats in the red box mean them in the heat-stressed period, in which the environmental temperature humidity index (THI) was 86.80, and goats in this environment for 14 days.

**Figure 2 animals-12-02455-f002:**
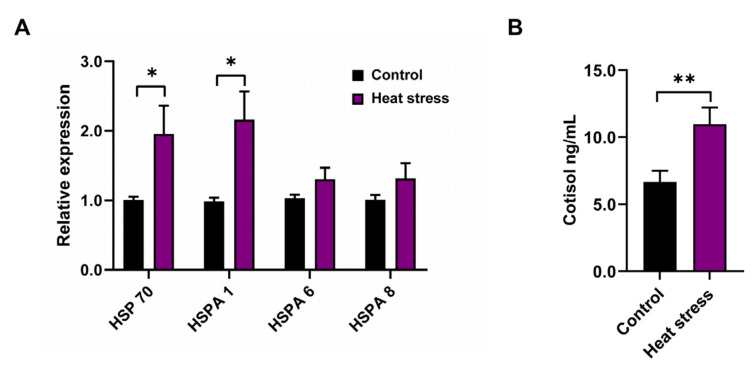
The indicators to evaluate the occurrence of heat stress in goats. (**A**) The expression levels of Hsp 70 family member genes in blood and rumen fluids (*n* = 6). (**B**) The serum cortisol concentrations in goats in the control and heat stress periods (*n* = 12). Data are expressed as the means ± SEM. * *p* < 0.05 and ** *p* < 0.01, indicating significant differences between the control and heat stress periods.

**Table 1 animals-12-02455-t001:** The diet compositions and nutritional levels (g/kg) of the basal diet of goats.

Ingredient	Content
Alfalfa	552
Ground corn	274
Soybean meal	74
Wheat barn	63
Ca_2_HPO_4_	7
Premix *	10
**Nutrition Level**	
Dry matter	946
Organic matter	858
Crude protein	177
Neutral detergent fiber	435
Acid detergent fiber	260
Ca	5.9
P	3.2

* Premix contained per kg: 20.70 g Mg, 0.50 g Fe, 1 g Mn, 2 g Zn, 43 mg Se, 47 mg I, 54 mg Co, 90,000 IU vitamin A, 17,000 IU vitamin D, 1750 IU vitamin E.

**Table 2 animals-12-02455-t002:** The details of primer sequences of HSP 70 genes.

Gene	Primer Sequence	Product Length	Annealing Temperature	GenBank Accession No.
B-actin	F: TCTGGCACCACACCTTCTAC R: TCTTCTCACGGTTGGGCCTTG	102	60	XM 018039831.1
HSP 70	F: TGGCTTTCACCGATACCGAG R: GTCGTTGATCACGCGGAAAG	167	60	NM 001285703.1
HSPA 1	F: CGACCAGGGAAACCGGCAC R: CGGGTCGCCGAACTTGC	151	60	NM 005677146.3
HSPA 6	F: TCTGCCGCAACAGGATAAA R: CGCCCACGCACGAGTAC	239	60	NM_001314233.1
HSPA 8	F: ACCTCTATTACCCGTGCCC R: CTCTTATTCAGTTCCTTCCCATT	203	60	XM 018039831.1

**Table 3 animals-12-02455-t003:** The skin temperature, rectal temperature, respiratory rate, and pulse of the goats between control and heat stress periods.

	Periods			
**Parameters**	**Control**	**Heat Stress**	**SEM**	***p* Values**
Rectal temperature (°C)	39.51 ^a^	39.44 ^b^	0.11	0.232
Skin temperature (°C)	33.11 ^a^	36.54 ^b^	1.23	0.047
Pulse (beats/min)	76.30 ^a^	85.12 ^b^	6.53	0.002
Respiratory rate (breaths/min)	25.43 ^a^	33.1 ^b^	4.25	0.032

The different lowercase superscripts in the rows indicate significant differences (*p* < 0.05); the same lowercase superscripts in the rows indicate no differences (*p* > 0.05).

**Table 4 animals-12-02455-t004:** The parameters of rumen fermentation and the activities of ruminal cellulolytic enzymes of heat-stressed goats with probiotic supplements.

	Treatment				SEM	*p* Value		
Parameters	Control	CB	SC	Mix		SC	CB	Mix
pH	6.54 ^a^	6.87 ^b^	6.80 ^b^	6.83 ^b^	0.17	<0.001	<0.001	<0.001
ORP (mV)	−161.3 ^a^	−191.0 ^b^	−193.4 ^b^	−197.1 ^b^	8.24	0.044	<0.001	0.022
NH_3_-N (mg 100 mL^−1^)	9.17 ^a^	10.89 ^ab^	12.23 ^b^	13.81 ^b^	1.47	0.041	0.021	0.041
TVFA (mmol L^−1^)	32.84 ^a^	51.22 ^b^	52.68 ^b^	52.98 ^b^	6.45	0.043	<0.001	<0.001
Acetic acid (mmol L^−1^)	14.38 ^a^	24.12 ^b^	24.77 ^b^	25.59 ^b^	3.08	0.009	<0.001	<0.001
Propionic acid (mmol L^−1^)	10.08 ^a^	15.22 ^b^	16.24 ^b^	15.73 ^b^	3.44	0.004	0.023	<0.001
Butyric acid (mmol L^−1^)	8.38 ^a^	11.90 ^b^	11.67 ^b^	11.69 ^b^	2.68	0.044	0.040	0.056
A/P ratio	1.43 ^a^	1.59 ^b^	1.53 ^b^	1.63 ^b^	0.55	0.007	0.049	0.022
Avicelase (IU mL^−1^)	1.30 ^a^	1.56 ^b^	1.61 ^b^	1.81 ^b^	0.02	0.040	<0.001	<0.001
CMCase (IU mL−^1^)	1.34 ^a^	2.51 ^b^	2.58 ^c^	3.12 ^b^	0.01	<0.001	<0.001	<0.001
Cellobiase (IU mL^−1^)	2.44 ^a^	4.43 ^b^	4.51 ^b^	4.73 ^b^	0.05	<0.001	<0.001	<0.001
Xylanase (IU mL^−1^)	4.54 ^a^	6.43 ^b^	7.21 ^c^	7.62 ^b^	0.10	<0.001	<0.021	0.043

The different lowercase superscripts in the rows indicate significant differences (*p* < 0.05); the same lowercase superscripts in the rows indicate no differences (*p* > 0.05).

**Table 5 animals-12-02455-t005:** The parameters of growth performance in heat-stressed goats with probiotic supplementation.

	Groups					*p* Value		
Parameters	Control	CB	SC	Mix	SEM	SC	CB	Mix
DMI (kg)	0.70 ^a^	0.85 ^b^	0.85 ^c^	0.88 ^b^	0.12	0.045	<0.001	<0.001
ADG (kg)	0.08 ^a^	0.15 ^b^	0.17 ^b^	0.19 ^b^	0.04	0.005	<0.001	0.004
Digestibility of								
DM (%)	51.57 ^a^	59.48 ^b^	60.64 ^c^	63.34 ^b^	6.63	0.002	<0.001	<0.001
NDF (%)	39.23 ^a^	47.40 ^b^	50.31 ^b^	51.02 ^b^	3.09	<0.001	<0.001	0.040
ADF (%)	35.28 ^a^	45.30 ^b^	48.60 ^b^	49.29 ^b^	3.47	<0.001	<0.001	<0.001

The different lowercase superscripts in the rows indicate significant differences (*p* < 0.05); the same lowercase superscripts in the rows indicate no differences (*p* > 0.05).

## Data Availability

Not applicable.

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
