# Peer review of "Prophylactic Feeding of Clostridium butyricum and Saccharomyces cerevisiae Were Advantageous in Resisting the Adverse Effects of Heat Stress on Rumen Fermentation and Growth Performance in Goats"

_animals, 2022, doi:10.3390/ani12182455_

Round 1
Reviewer 1 Report
A fascinating manuscript with practical implications, but some issues need to be explained:
1. Are 14 days sufficient for establishing heat stressed goat model?
2. How are the supplemented levels of Clostridium butyrate and Saccharomyces
cerevisiae determined?
3. Please add the method of ORP measure in the " Sampling and measurements” section and explain why you choose this index for the rumen environment.
4. At line 164, please add the package of the R studio.
5. In the "results” section, please add the result of ORP.
6. At line 220, change “In the hot season...” into “In the hot summer...”
7. At line 302,“ [Soren]” is not the right way to cite documents.
Author Response
Response to Reviewer 1 Comments
Dear reviewer,
Thanks for your thoughtful comments and suggestions. As shown in the manuscript, we have made careful modifications to the manuscript. We have made thorough revisions to improve the English as needed. Below you will find our general reply and point-by-point responses to the reviewers’ comments/ questions.
Point 1: Are 14 days sufficient for establishing heat stressed goat model?
Response 1: Thank you for your question. That is enough! Because during 14 days, the THI was greater than 82. Moreover, the expression levels of HSP70 family member genes significantly increased in blood and rumen fluid. Besides, the serum cortisol concentration significantly increased. All the parameters indicated the occurrence of heat stress. Moreover, based on our previous studies, the heat stress goat model was successfully established for 14 days.
Point 2: How are the supplemented levels of Clostridium butyrate and Saccharomyces cerevisiae determined?
Response 2: According to the results of our previous studies, the supplemented levels of Clostridium butyrate and Saccharomyces cerevisiae at 0.05 % and 0.6%, respectively, were the optimal supplemented levels for improving rumen fermentation and growth performance of heat-stressed goats.
Point 3: Please add the method of ORP measure in the “Sampling and measurements” section and explain why you choose this index for the rumen environment.
Response 3: ORP is an indicator that indirectly reflects the oxygen content in the rumen. When heat stress occurs, the ruminal ORP significantly increases due to the increased respiration rate, which causes the rumen to contain more O2. Moreover, the increased frequency and duration of water intake at high environmental temperatures also contribute to the increase in O2. The rise in ORP may induce adverse effects on anaerobic microbiota in the rumen, resulting in a decrease in their abundance and/or activity. The method of ORP measure was added at lines 143-145.
Point 4: At line 164, please add the package of the R studio.
Response 4: The “stats” package of the R studio has been added at line 166.
Point 5: In the “results” section, please add the result of ORP.
Response 5: the result of the effects of supplementation with Clostridium butyrate, Saccharomyces cerevisiae, and their mixture on ORP have been added at lines 198-200.
Point 6: At line 220, change “In the hot season...” into “In the hot summer...”
Response 6: “In the hot season...” has been changed into “In the hot summer...” at line 222.
Point 7: At line 302,“ [Soren]” is not the right way to cite documents.
Response 7: The cite “ [Soren]” has been changed to the right way to cite as [52] at line 304.
Reviewer 2 Report
Line 88: it is suggested to include a basis for calculating the amount of basal diet
Line 89: Suggests concentrator to include the proportion of the diet
Line 89: It is suggested to include individual animals in collective pens or 89:
Line 90 to 92: It is suggested to include the basis for calculating the additives
Line 113: It is suggested to include the sample collection method and samples obtained by samples.
Line 18: The performance has barely been completed. Also described are feed conversion data, average daily gain.
Line 150: Why didn't you do in situ digestibility analysis?
Line 155: The topic starts with an incorrect numbering. According to the order it should be 2.3
Line 156: Why not use the SAS program?
Line 158: Why was this test used? It was not clear the statistics used to analyze the data on the interaction between additives and temperature.
Line 172: review the form of writing, as other problems related to heat stress did not show statistical difference.
Line 209: The topic starts with an incorrect numbering. According to the order it should be 4.
Author Response
Dear reviewer,
Thanks for your thoughtful comments and suggestions. As shown in the manuscript, we have made careful modifications to the manuscript. We have made thorough revisions to improve the English as needed. Below you will find our general reply and point-by-point responses to the reviewers’ comments/ questions.
Point 1: Line 88: it is suggested to include a basis for calculating the amount of basal diet
Response 1: Thank you for your suggestion. The basis for calculating the amount of basal diet was according to Mo, 2011, and this reference has been added at lines 92-93.
Point 2: Line 89: Suggests concentrator to include the proportion of the diet
Response 2: Thank you for your suggestion. In our opinion, the existing feed formula and expression of nutrient levels are well understood by the reader, so we do not make any adjustments to it.
Point 3: Line 89: It is suggested to include individual animals in collective pens or 89:
Response 3: Thank you for your suggestion. In this study, each goat was kept in a separate pen for sampling convenience.
Point 4: Line 90 to 92: It is suggested to include the basis for calculating the additives
Response 4: The basis for calculating the additives was according to our previous studies that the supplemented levels of Clostridium butyrate and Saccharomyces cerevisiae at 0.05 % and 0.6%, respectively, were the optimal supplemented levels for improving rumen fermentation and growth performance of heat-stressed goats.
Point 5: Line 113: It is suggested to include the sample collection method and samples obtained by samples.
Response 5: The collection method of fresh fecal, blood, and rumen fluid samples were listed at lines 117-118, 119-120, and 141-143, respectively.
Point 6: Line 18: The performance has barely been completed. Also described are feed conversion data, average daily gain.
Response 6: The study was an exploratory study, and only DMI, ADG, and the digestibilities of DM, NDF, and ADF were taken into consideration as growth performance parameters. We will add the indicators that you suggested in our future studies.
Point 7: Line 150: Why didn't you do in situ digestibility analysis?
Response 7: The method we have adopted for digestibility analysis can accurately analyze the feed digestibility. This method has been used in several of our previous studies, and we are familiar with the operation method, so we did not adopt in situ digestibility analysis.
Point 8: Line 155: The topic starts with an incorrect numbering. According to the order it should be 2.3
Response 8: The incorrect numbering has been revised as “2.3” at line 165.
Point 9: Line 156: Why not use the SAS program?
Response 9: The R language is a more powerful tool for scientific research. Besides statistics, many other issues in scientific research can be solved by R language.
Point 10: Line 158: Why was this test used? It was not clear the statistics used to analyze the data on the interaction between additives and temperature.
Response 10: This test was used to analyze DM, NDF, and ADF in feed and feces and calculate the digestibilities of these paraameters. It is not necessary to analyze the interaction between additives and temperature. After successful modeling of heat-stressed goats, the effects of probiotics on rumen fermentation and growth performance be compared among groups.
Point 11: Line 172: review the form of writing, as other problems related to heat stress did not show statistical difference.
Response 11: Through the full-text examination, we marked the difference in heat stress parameters.
Point 12: Line 209: The topic starts with an incorrect numbering. According to the order it should be 4.
Response 12: By double checking, the whole text and incorrect numbering of the tables were revised at lines 209-210, 219-220.
Round 2
Reviewer 2 Report
Thank you for making the suggested changes.